# Bioengineered textiles with peptide binders that capture SARS-CoV-2 viral particles

Laura Navone [1,2 ✉], Kaylee Moffitt[1], Wayne A. Johnston[1,2], Tim Mercer[3,4], Crystal Cooper[5], Kirsten Spann[6] & Robert E. Speight [1,2]

The use of personal protective equipment (PPE), face masks and ventilation are key strategies to control the transmission of respiratory viruses. However, most PPE provides physical protection that only partially prevents the transmission of viral particles. Here, we develop textiles with integrated peptide binders that capture viral particles. We fuse peptides capable of binding the receptor domain of the spike protein on the SARS-CoV-2 capsid to the cellulose-binding domain from the *Trichoderma reesei* cellobiohydrolase II protein. The hybrid peptides can be attached to the cellulose fibres in cotton and capture SARS-CoV-2 viral particles with high affinity. The resulting bioengineered cotton captures 114,000 infective virus particles per $cm^2$ and reduces onwards SARS-CoV-2 infection of cells by 500-fold. The hybrid peptides could be easily modified to capture and control the spread of other infectious pathogens or for attachment to different materials. We anticipate the use of bioengineered protective textiles in PPE, facemasks, ventilation, and furnishings will provide additional protection to the airborne or fomite transmission of viruses.

[1] School of Biology and Environmental Sciences, Faculty of Science, Queensland University of Technology (QUT), Brisbane, QLD 4000, Australia. [2] ARC Centre of Excellence in Synthetic Biology, Queensland University of Technology (QUT), Brisbane, QLD 4000, Australia. [3] Australian Institute for Bioengineering and Nanotechnology, The University of Queensland (UQ), Brisbane, QLD 4072, Australia. [4] Garvan Institute of Medical Research, Sydney, NSW 2010, Australia. [5] Central Analytical Research Facility (CARF), Queensland University of Technology (QUT), Brisbane, QLD 4000, Australia. [6] Centre for Immunology and Infection Control, School of Biomedical Science, Faculty of Health, Queensland University of Technology (QUT), Brisbane, QLD 4000, Australia. ✉email: laura.navone@qut.edu.au

The transmission of viruses can occur by the airborne spread of aerosols containing viral particles. Saliva and respiratory droplets containing viral particles are expelled by coughing, sneezing or speaking of an infected host. These airborne viral particles are then inhaled and cause infection in new hosts. New hosts may be alternatively infected by fomite transmission, which occurs when a host directly contacts objects or surfaces contaminated with viral particles[1,2].

Severe acute respiratory syndrome coronavirus 2 (SARS-CoV-2) that causes COVID-19 is primarily transmitted through aerosols and, less commonly, by fomite transmission[1,2]. Public health measures, such as physical distancing, ventilation and wearing face masks, have been widely implemented to limit the spread of the virus. However, further innovations are required to improve virus containment and reduce infection[3,4].

The use of personal protective equipment (PPE), such as face masks, has proven an effective approach to prevent exposure and transmission of infectious pathogens[5–7]. The effectiveness of face masks to limit the spread of aerosols containing viral particles depends on the material's filtration efficacy and the design's fit[5,8]. Respirator masks (i.e., KN95, N95, N99 and FFP1-3) show ~95% aerosol filtration efficacy, while the effectiveness of cloth masks varies widely from 12 to 99.9%[9]. If viral particles are not captured and filtered by face masks, a new host can become infected.

Synthetic biology has enabled the development of engineered biomolecules capable of detecting exposure to infectious pathogens that can be attached to textiles and wearables[10]. Recently, textiles able to detect contact with infectious pathogens by synthetic RNA circuits were developed. However, these textiles only indicate viral presence and are not able to capture infectious particles and prevent onwards transmission. Nanofibers with antiviral coatings derived from copper and silver nanoparticles have been developed to improve the protection of PPE against SARS-CoV-2[11,12]. Yet, these antiviral coatings expose individuals to heavy metals that may lead to health issues, such as dermatological reactions[13,14]. The design of non-toxic biomolecules such as hybrid proteins that can bind and capture virus particles onto textiles fibres would represent a benign alternative to these approaches.

SARS-CoV-2 infects cells by binding the angiotensin-converting enzyme (ACE2) receptor on the host cell with the viral spike protein receptor-binding domain (RBD)[15]. Peptide inhibitors that bind the spike protein with high affinity can directly compete with ACE2 binding and neutralise SARS-CoV-2 infection in cell culture. These anti-RBD binders, named AHB2 and LCB1, were designed by de novo computational modelling as potential therapeutics, and their effectiveness has been demonstrated in animal models[16,17]. Previous in vitro studies of anti-RBD binders showed high binding affinity to SARS-CoV-2 RBD ($K_D$ of 15.5 nM and 625 pM for AHB2 and LCB1, respectively)[16,17].

Here, we developed a new class of hybrid peptides that fuse anti-RBD binders with Cellulose Binding Domains (CBDs) able to bind cellulose fibres in textiles. These hybrid capture peptides are attached to cotton to generate bioengineered textiles capable of capturing SARS-CoV-2 (Fig. 1). We show how these textiles bind viral particles with high affinity and prevent onwards infection of mammalian cells. The bioengineered textiles can be used in face masks and ventilation systems to trap and neutralise airborne viral particles or used in clothing and furnishing to limit fomite transmission. These bioengineered peptides can extend the protective capabilities of common materials to onwards viral transmission.

## Results

### Design of fusion peptides that capture viral particles and attach to materials.
We first wanted to develop a hybrid 'capture' peptide capable of binding viral particles that could also be attached to cotton. For this, we used the recently developed AHB2 and LCB1 peptides that bind the RBD domains of the spike protein on the outer capsid of SARS-CoV-2[16,17]. AHB2 was designed using a Rosetta blueprint builder based on the ACE2 alpha-helix that makes the majority of contacts with the RBD protein[16]. LCB1 was generated de novo by designing binders using rotamer interaction field (RIF) docking to distinct regions of the RBD surface surrounding the ACE2 binding site[16,17]. LCB1 has been shown to protect mice (expressing human ACE2) against infection by the B.1.17 variant (alpha), B.1.351 (beta) and B.1.1.28 variants[17]. Given this broad protection against several SARS-CoV-2 strains, we considered *LCB1* a promising candidate for our designs.

We next selected protein domains capable of binding cellulose fibres[18]. We used characterised CBDs from cellobiohydrolase II CH2 from *Trichoderma reesei* and cellobiohydrolase CEX from *Cellulomonas fimi* which have demonstrated binding capabilities to crystalline cellulose and cotton fibres[18,19]. We also selected an amino acid linker that is resistant to proteolysis in the yeast *Pichia pastoris*[20] to fuse the anti-RBD and CBD sequences (Fig. 1). Using this approach, we engineered four capture peptides called AHB2-CH2, LCB1-CH2, AHB2-CEX and LCB1-CEX according to their component domains (complete construct sequences can be found in Supplementary Data 1).

### Capture peptides are compatible with industrial biomanufacture.
The small size of the capture peptides (~15 kDa) facilitates production in the methylotrophic yeast platform, *Pichia pastoris* that is extensively used in industry for the manufacture of recombinant proteins. *P. pastoris* has a high secretory capability for heterologous proteins, while secreting low amounts of endogenous proteins. This simplifies otherwise costly downstream processing, and achieves low production costs and facile scale-up[21]. However, engineering of *P. pastoris* strains to achieve high productivity requires extensive optimisation[22].

We designed the genetic elements to enable extracellular secretion of the capture peptides using the α-mating-factor sequence from *Saccharomyces cerevisiae*. We also chose a strong bidirectional promoter, $P_{HpFMD-HpMOX}$ from *Hansenula polymorfa*[21,23], to drive co-expression of protein disulfide isomerase (PDI), which improves the production of recombinant proteins that contain disulfide bonds[22], such as the CEX and CH2 domains that include one and three disulfide bonds in their amino acid sequence, respectively[19,22] (Fig. 1 and Supplementary Fig. 1).

We successfully produced all capture peptides in flask fermentations at yields compatible with industrial process development (~250–400 μg/mL in shake flasks). The capture peptides were separated from yeast cells by centrifugation followed by filtration and used for binding experiments without the need for purification. We also generated the non-fused anti-RBD binders, AHB2 and LCB1, without CBD domains as control for binding experiments to cellulose. Furthermore, we also treated the capture peptides with Endo H glycosidase and confirmed they were not glycosylated on SDS-PAGE. The incorporation of glycan chains during post-translational processing in eukaryotic systems could influence the binding of capture peptides to materials or to the spike protein RBD[24,25]. Overall, these results demonstrate the production and secretion of capture peptides *P. pastoris* yeast cells at yields compatible with industrial process development.

### Attachment of capture peptides to textiles.
CBDs like CH2 and CEX typically achieve very high-affinity binding to cellulose chains due to strong interactions between aromatic amino acids on the hydrophobic surface of cellulose, as well as hydrogen and Van der Waals interactions between the hydroxyl of the

## Manufacture of bioengineered textiles.

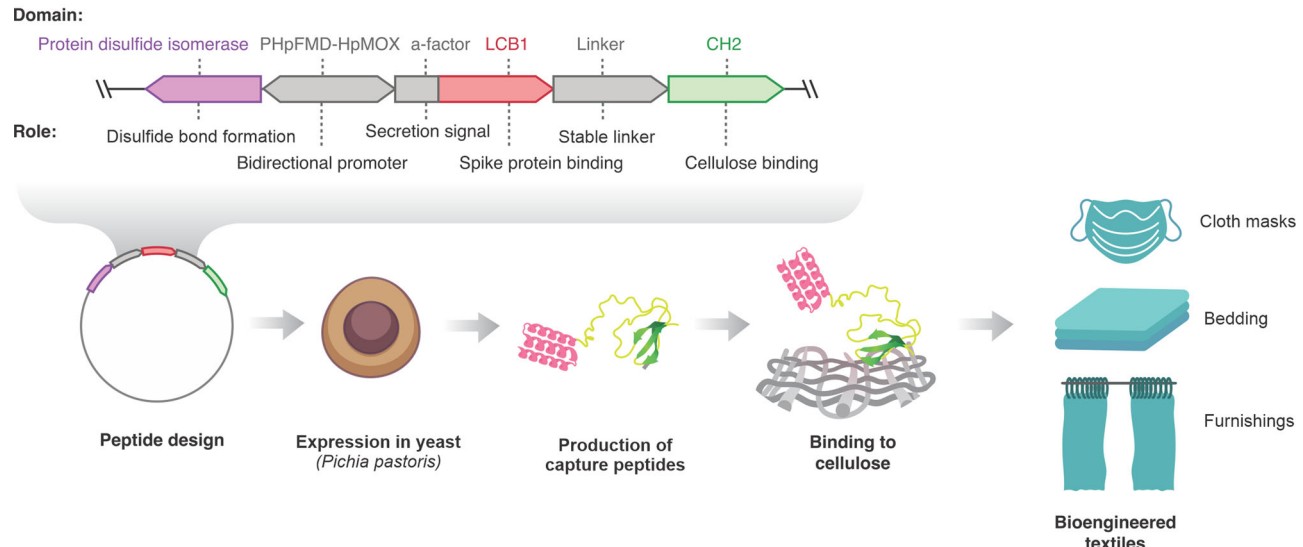

**Fig. 1 Manufacture of bioengineered textiles.** Schematics of the genetic design and production of capture peptides in *Pichia pastoris* fermentations. We developed hybrid capture peptides capable of binding viral particles that can also be attached to cellulose. We used AHB2 and LCB1 peptides that bind the RBD domains of the SARS-CoV-2 spike protein and fused them to protein domains capable of binding cellulose fibres (CBDs). The α-mating-factor sequence enables extracellular secretion of the capture peptides while the strong bidirectional promoter, $P_{HpFMD-HpMOX}$, drives co-expression of protein disulfide isomerase (PDI) to improve production of CBDs that contain disulfide bonds in their structure.

glucopyranosyl ring and polar amino acids[26–29]. We tested the binding of the capture peptides to two cellulosic fabric types, cotton and rayon, containing different percentages of crystalline cellulose, 90% and 75%, respectively[30]. We also evaluated binding to Avicel PH101 microcrystalline cellulose, which is a common binding substrate for CBDs[31].

To evaluate the binding of the capture peptides to cotton, we incubated the peptides with cellulosic materials and measured the decrease in peptide concentration in the solution. Capture peptides AHB2-CH2 and LCB1-CH2 showed mean binding of 44% and 43% respectively to cotton fabric, 56% and 35% respectively to rayon fabric, and 60% and 49% respectively to Avicel (Fig. 2a, b). We observed a significant difference in binding between the CH2 and CEX capture peptides to cotton, rayon and Avicel, with CH2 presenting a higher affinity than CEX to all materials ($p < 0.01$). For this reason, further studies were conducted with CH2 capture peptides only. The AHB2 or LCB1 peptide controls without CBD fusions showed no binding (0%) to materials (not included in Fig. 2b).

We next performed isothermal absorption assays to analyse the binding affinity of the capture peptide AHB2-CH2 to cotton, rayon and Avicel at pH 5 and 4 °C, using a Langmuir adsorption isotherm model fit ($p < 0.001$) (see Supplementary Fig. 2a). Dissociation constants ($K_d$) for cotton, rayon and Avicel were $60.7 \pm 23.8$, $61.3 \pm 14.4$ and $92.0 \pm 38.2$ pM, respectively. This result confirmed the high-affinity binding of CH2 to cellulosic surfaces (see Supplementary Fig. 2b). We also measured the binding of AHB2-CH2 to cotton, rayon and Avicel to test the binding at pH 6 and 7, compared to pH 5. For cotton and rayon, binding did not improve at pH 6, and binding and significatively decreased at pH 7 ($p < 0.01$) (see Supplementary Fig. 2c).

We next evaluated the time required to attach the capture peptides to cotton for manufacture and whether the bioengineered cotton retained the capture peptides washing conditions. We performed binding of capture peptides to cellulosic materials for 2, 4 or 16 h at room temperature and 4 °C. We found no significant difference in binding of the capture peptides at 16 h at

room temperature or at 4 °C (see Supplementary Fig. 2d, e). Results also showed that mean binding of the capture peptides at 4 h at room temperature was not significantly different compared to 16 h binding. However, mean binding for 4 h at 4 °C was significantly different to 16 h. This difference most likely dependent on variations in absorption kinetics at different temperatures for CH2 domain[32]. Overall, the results support the idea that the capture peptides can be easily attached to textiles, such as during routine laundry cycles[33,34].

The washing or wetting of textiles may denature and remove attached capture peptides. To measure the impact of wetting and washing on the protective performance of bioengineered textiles, we treated the bioengineered textiles with different buffers, including phosphate-buffered saline (PBS) and phosphate buffer pH 5, media Dulbecco's Modified Eagle Medium (DMEM), as well as deionised water. We did not observe an increase in peptide concentration in solution after performing the washes (see Supplementary Fig. 2f). Conversely, to confirm whether capture peptides were still bound to the materials after the buffer and media washes, we treated the materials with 1% SDS. If the capture peptides were still attached to the cellulose fibres after the consecutive washes, the treatment with SDS detergent would denature and remove them from the material's surface. We were able to recover 95 to 100% of capture peptides after SDS treatment as determined by protein concentration[35]. Together, these results confirm that bioengineered materials can be soaked with buffer, media, or water with relatively little loss of capture peptide, suggesting that moisture in breath, sneezing and other respiratory droplets would not release the capture peptides from the material. However, washing textiles with detergent (i.e., SDS) can remove the capture peptides.

**Bioengineered textile specifically binds GFP-RBD hybrid protein.** We next tested the capability of the bioengineered cotton to bind the SARS-CoV-2 spike protein. We designed an in vitro pull-down experiment wherein the spike protein RBD domain

**a.** Design of capture peptides and structural models.

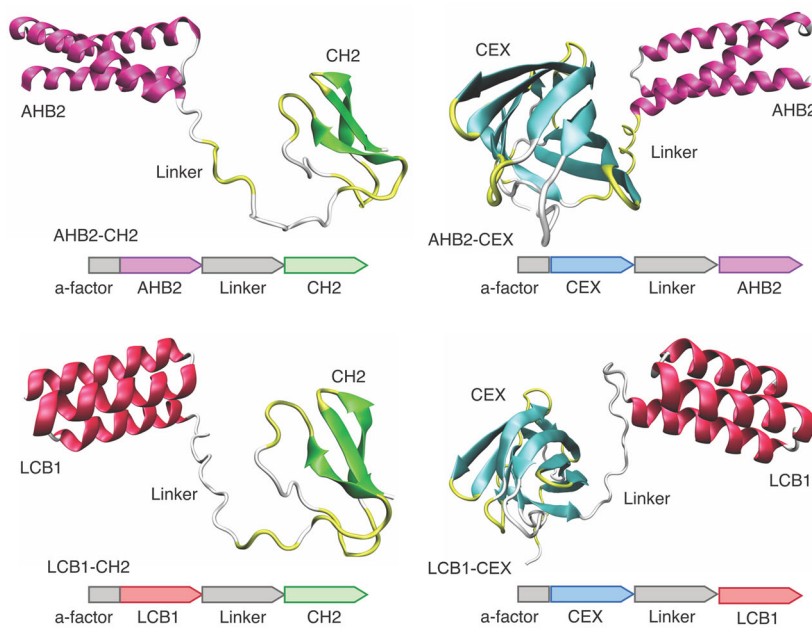

**b.** Binding of capture peptides to cellulose.

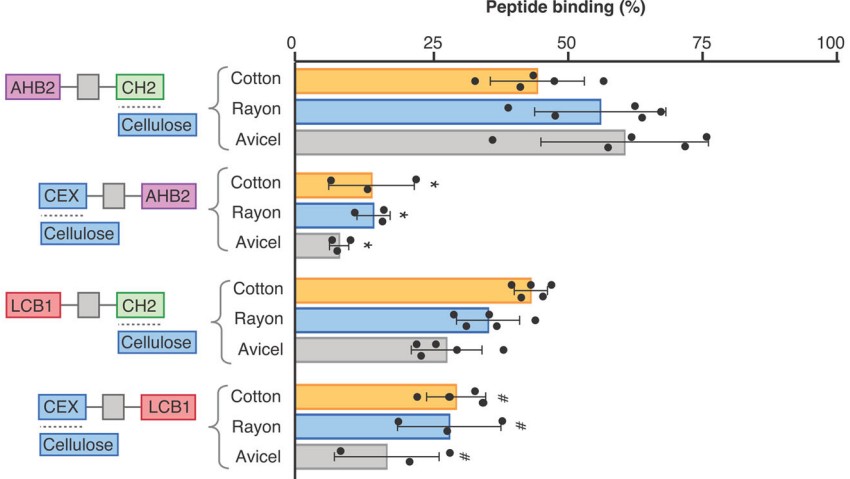

**Fig. 2 Design of capture peptides and binding to cellulose.** We incubated capture peptides with cellulosic materials and determined the decrease of peptide concentration in solution. **a** Structural models and genetic design of capture peptides AHB2-CH2, LCB1-CH2, AHB2-CEX and LCB1-CEX. Models were generated using CoLab: AlphaFold2 notebook[53]. **b** Binding of capture peptides to cellulosic materials, cotton, rayon and Avicel at pH 5 and 4 °C. Data are presented as mean and standard deviation of biological replicates and analysed using one-way ANOVA with multiple comparisons. Filled black circles show individual data points. *Significantly different to AHB2-CH2 binding to cotton at pH 5 ($p < 0.01$). #Significantly different to LCB1-CH2 binding to cotton at pH 5 ($p < 0.01$).

was fused to a GFP reporter (GFP-RBD) to enable the relative capture of the spike protein to be measured by fluorescence (Fig. 3a). A *Leishmania* based cell-free protein expression system was used to produce the GFP-RBD reporter (see "Methods").

To evaluate the capture of the SARS-CoV-2, we incubated the bioengineered AHB2-CH2 and LCB1-CH2 cotton in GFP-RBD solution. We then measured the decrease in relative fluorescent units (RFUs) in the solution following a 1-h incubation at 30 °C. GFP alone was also produced in the cell-free system and included as a negative control (Fig. 3b). We found that bioengineered cotton with capture peptides demonstrated 48% and 33% binding efficacy, respectively, to GFP-RBD (Fig. 3a). We observed no significant differences between GFP-RBD binding to either of the capture

peptides tested ($p < 0.01$). To further visualise the binding of GFP-RBD to the bioengineered textile, we analysed the samples with confocal fluorescence microscopy. We observed strong fluorescence on the bioengineered cotton fibres, showing the capture peptides specifically attached to the GFP-RBD reporter (Fig. 3a, b).

We next evaluated whether dry bioengineered textiles could capture SARS-CoV-2 RBD under conditions similar to when respiratory droplets with viral particles might encounter dry textiles. We performed an in vitro binding experiment with GFP-RBD reporter using completely dried bioengineered cotton. After capture peptide binding, the dried textile was re-hydrated using saline solution and demonstrated 40% and 26% binding efficacy for AHB2-CH2 and LCB1-CH2 respectively, to GFP-RBD

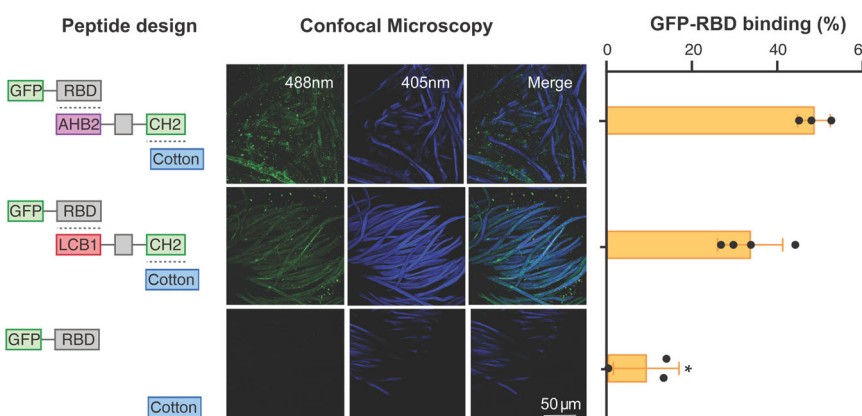

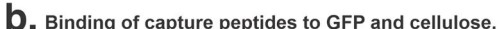

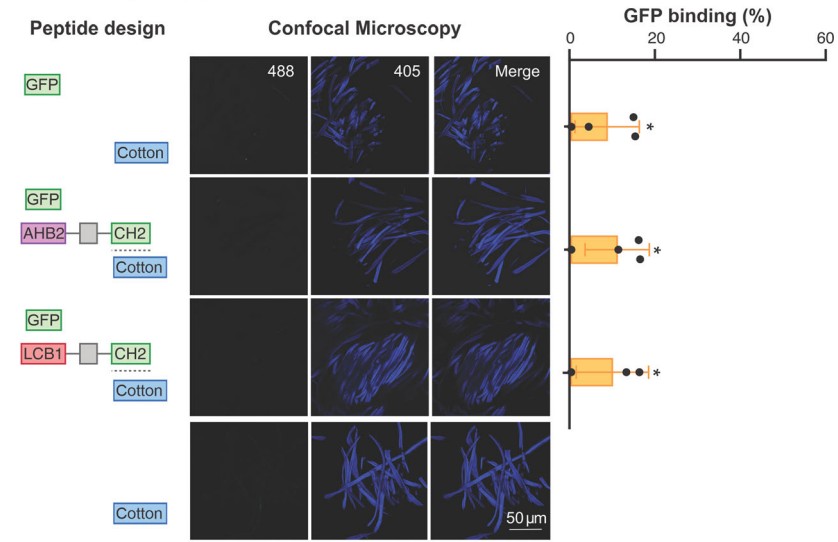

**Fig. 3 Binding of bioengineering textiles to GFP-(SARS-CoV-2 Receptor Binding Domain).** We studied the capability of the bioengineered cotton to bind the SARS-CoV-2 spike protein using a GFP reporter fusion to the RBD and measured the capture of the spike protein domain by decrease in fluorescence. **a** Binding of capture peptides to GFP-RBD or (**b**) GFP and cellulose. Confocal fluorescence microscopy of GFP-RBD or GFP binding to bioengineered cotton with capture peptides AHB2-CH2 or LCB1-CH2. Percentage of GFP-RBD or GFP binding bioengineered cotton was calculated from the decrease in relative fluorescent units (RFUs). Cotton shows autofluorescence from the 405 nm excitation wavelength that does not interfere with GFP fluorescence at 488 nm. Data are presented as the mean and standard deviation of biological replicates and analysed using one-way ANOVA with multiple comparisons. Filled black circles show individual data points. *Significantly different to GFP-RBD/AHB2-CH2 binding ($p < 0.001$).

with no significant decrease from previous binding results (see Supplementary Fig. 3).

**Bioengineered cotton neutralises SARS-CoV-2 infection.** We finally investigated the protective capability of the bioengineered cotton to SARS-CoV-2 infection. To evaluate this, we measured whether the sequestration of the viral particles could reduce the onward infection of cell cultures. The bioengineered cotton was incubated with a suspension of $6.8 \times 10^4$ particle-forming units (PFUs) of SARS-CoV-2 particles for 1 h at room temperature. The cotton was then washed with media to recover the unbound virus. We then prepared 10-fold serial dilutions of the wash and infected Vero cell monolayers (Fig. 4a). The infectivity of the virus on the cell culture was then measured using a TCID50/mL (Median Tissue Culture Infectious Dose) assay after a four-day incubation at 37 °C (see "Methods").

We observed the bioengineered cotton markedly reduced the SARS-CoV-2 TCID50/mL that was recovered, by 139 and 146-

fold for AHB2-CH2 and LCB1-CH2, respectively, compared to the unbound cotton control, and by 470 and 500-fold for AHB2-CH2 and LCB1-CH2, respectively, compared to virus control ($p < 0.001$) (Fig. 4b and Supplementary Fig. 4). We also calculated recovered PFUs from the TCID50/mL assay and compared them to the initial PFUs of SARS-CoV-2 suspension used in the experiment. From this calculation we report 114,000 PFUs captured per cm² of cotton for both AHB2-CH2 and LCB1-CH2 bioengineered textile. This confirms that the bioengineered cotton sequestered the SARSCoV-2 viral particles and reduced infection.

The AHB2 and LCB1 anti-RBD binders were designed to neutralise further infection by high-affinity binding to the viral spike protein, as shown in previous reports for AHB2 and LCB1[16,17]. We confirmed this inhibitory effect by incubating SARS-CoV-2 with the capture peptides, AHB2-CH2 and LCB1-CH2, in solution rather than bound to cotton. The infectivity measured by TCID50/mL of SARS-CoV-2 in Vero cells was significantly decreased by 160 and 610-fold for AHB2-CH2 and

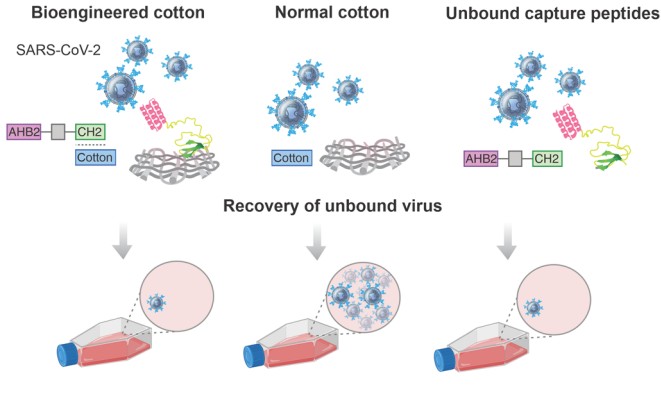

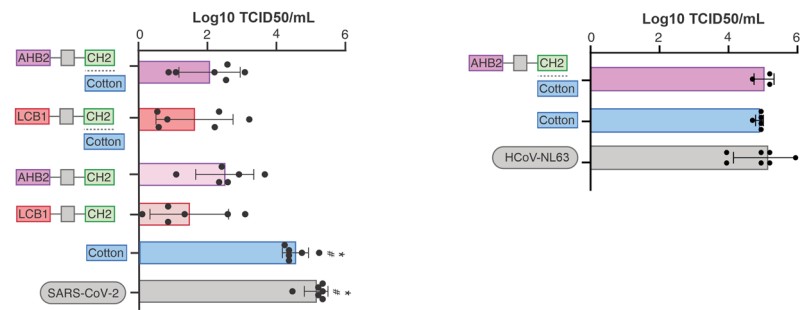

**Fig. 4 Sequestration of SARS-CoV-2 viral particles by bioengineered textiles. a** Schematics of sequestration of SARS-CoV-2 and infection of culture cells. Unbound virus was recovered and used to infect Vero cells monolayers. Our bioengineered textile neutralises and decreases the onwards transmission of SARS-CoV-2 in cell models. **b** Binding of SARS-CoV-2 to bioengineered cotton, unbound capture peptides and unbound normal cotton. Infectivity of recovered unbound virus shown as log10 TCID50/mL. SARS-CoV-2 control indicated as SARS-CoV-2. **c** Binding of HCoV-NL63 to bioengineered cotton and unbound normal cotton. Infectivity of recovered unbound virus shown as log10 TCID50/mL. HCoV-NL63 control indicated as HCoV-NL63. TCID50/mL was calculated using the Spearman–Kaerber algorithm. Data are shown as the mean and standard deviation of six biological replicates and analysed using one-way ANOVA with multiple comparisons. Individual data points are shown as black filled circles. *Significantly different to AHB2-CH2/Cotton and LCB1-CH2/Cotton ($p < 0.001$). #Significantly different to AHB2-CH2 and LCB1-CH2 ($p < 0.001$).

LCB1-CH2, respectively, compared to the SARS-CoV-2 control with no peptides ($p < 0.001$) (Fig. 4b). In contrast, SARS-CoV-2 remained recoverable and infectious from cotton alone with no significant drop in TCID50/mL recovered (3.4-fold) after an hour of exposure.

To evaluate the specificity with which our bioengineered cotton bound SARS-CoV-2 virus, we repeated the above in vitro infection experiment using a different human coronavirus NL63 strain, HCoV-NL63, which only shares 17% similarity with the SARS-CoV-2 RBD[36] despite also binding to the ACE2 cell receptor. We incubated bioengineered AHB2-CH2 cotton with HCoV-NL63 for 1 h and subsequently washed the cotton as described above with media, followed by infection of monolayers of LLC-MK2 with a 10-fold serial dilution of washes (Fig. 4c). We did not observe a significant decrease in the infection of the LLC-MK2 cells with HCoV-NL63 following incubation with capture peptide, either in solution or bound to cotton, as measured by the viral TCID50/mL. This result shows that the AHB2 binder is specific for SARS-CoV-2 and the bioengineered textile did not sequester the HCoV-NL63 onto the cotton surface. Alternative binders for HCoV-NL63 (or other viruses) would need to be fused to CBD CH2 to develop a bio-textile targeting these viruses.

The amount of virus expelled by an infected person depends on the severity of the infection and the intensity of coughs or sneezing. To estimate the capability of our material to protect against SARS-CoV-2 in this real-world scenario, we compared our results with a cough virus expulsion modelling study. This study estimated a person with a high viral load ($\sim 2.35 \times 10^9$ virus copies per mL in oral fluid) generates up to $1.23 \times 10^5$ virus copies per mL in fluid droplets[37,38]. This load is lower than the expected number of virus particles that our bioengineered material could sequester. In a real-world scenario, the bioengineered textiles which have a capturing capability at 114,000 PFU per $cm^2$ of cotton could capture viral particles without approaching saturation. This suggests that bioengineered cotton in cloth masks and other PPE could capture viral particles and prevent onward transmission.

## Discussion

Here we developed bioengineered textiles capable of capturing viral particles and providing protection against viral transmission. We designed hybrid peptides capable of binding the spike protein from SARS-CoV-2 capsid, as well as cellulose fibres. These capture peptides can be attached to textiles, that can be washed and used to manufacture protective equipment. The bioengineered textiles can sequester viral particles and reduce subsequent infection in a cell model. We propose such bioengineered textiles can be used in manufacture of face masks, furnishings, bedding and ventilation systems to capture viral particles and prevent onwards aerosol or fomite transmission.

In this study we focused on developing bioengineered cotton capable of binding SARS-CoV-2, given the imminent need to develop infection control strategies against COVID-19. Bioengineered cotton could increase the efficacy of cloth masks that are commonly used as a barrier to SARS-CoV-2 transmission worldwide. However, modifying the binding domain within the hybrid peptide, we could similarly target other infectious pathogens, such as *influenza* or *Ebola*, for capture. Similarly, many different capture peptides, each targeting different viruses could be simultaneously attached to textiles to provide broad protection against many pathogens for several applications.

We showed that bioengineered cotton can be readily soaked in water or buffer, however, washing the cotton with detergents removed the capture peptides. This would prevent capture peptide removal by respiratory droplets or moisture during breathing. Conversely, the capture peptides attached to virus particles could be removed during laundry washing cycles with detergents. The capture peptides could even be re-applied during laundry cycles, for example at the rinse fabric softener stage. Alternatively, engineered peptides textiles and materials could also be sprayed and re-applied to textile surfaces as needed.

The capture peptides attached to the cotton improve the probability that a facemask worn by an infectious person will better capture expelled virus and prevent inhalation of viral particles. Masks developed using this technology could also provide a protective barrier that lasts longer than plain cotton, which may become permeable to virus and ineffective within hours[9,39]. In the specific case of the bioengineered textile used for bedding and furnishings, abrasion mechanisms by prolonged contact with the body should be considered and addressed in future developments.

There is concern that textiles that capture viral particles will become contaminated, and virus subsequently transferred[39]. The anti-RBD binders used here have been developed as protective therapies, where they are designed to neutralise binding of the spike protein. Similarly, we demonstrate that viral particles are sequestered with high affinity, and do not readily transfer. Nevertheless, the bioengineered textiles could be also used as a filtering layer within two cotton layers to further reduce this concern[40]. In future enhancements, the bioengineered textile may also contain virus inactivation chemicals, like quaternary ammonium chloride compounds or other non-toxic nanocoatings (nanoworms), to achieve both capture and kill[41–44]. Otherwise, the virus could be fully inactivated during washing procedures after use[5,45].

## Conclusion

Bioengineered materials can sequester viral particles and prevent onwards transmission in a broad range of settings. For example, attaching the engineered peptides to filters in ventilation systems may actively remove viral particles from circulating air in planes, buses, hospitals, workplaces or schools (Fig. 5). Here, we developed capture peptides capable of attachment to cellulose fibres used in textiles like cotton, rayon, and Lyocell. However, by replacing the CBD domain, the capture peptides could be bound to other materials, such as polypropylene[46], polystyrene[47], and keratin (e.g., wool)[48]. These bioengineered materials may be used in the manufacture of furniture and other homeware to sequester viral particles and prevent fomite transmission. After future user testing in face masks, ventilation filters and other infection control scenarios, the capture capabilities could also be combined with sensing approaches to provide new solutions to limit pathogen transmission throughout the community.

## Methods

**Strains and growth conditions**. The strain used in this study was the *P. pastoris* BG11 strain (derivative of *P. pastoris* BG10 strain, ΔAOX1 (mutS-methanol utilisation slow) from ATUM Inc. (Newark, California, USA). α-Select Silver efficiency

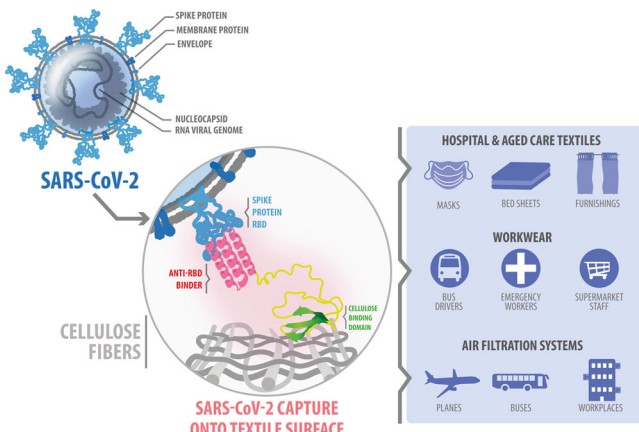

**Fig. 5 Applications of the bioengineered textile capture technology.** We focused on developing bioengineered cotton capable of binding SARS-CoV-2 to increase the efficacy of cloth masks commonly used as a barrier to COVID-19 transmission worldwide. By modifying the binding domain within the hybrid peptide, the textile technology can be used to target other infectious pathogens, such as *influenza* or *Ebola*, for capture. Extended applications of the bioengineered textiles include air filtration systems, beddings, and furnishings, providing additional protection by sequestering viral particles and limiting pathogen transmission throughout the community.

competent *E. coli* strain (Bioline, Australia) was used for cloning. Cultivations were conducted in Luria Broth (LB) media for *E. coli* and yeast cultures were either grown in YPD medium (1% w/v yeast extract, 2% w/v peptone and 2% w/v glucose), buffered minimal dextrose (BMD) medium (1.34% Yeast Nitrogen Base YNB, $4 \times 10^{-5}$% biotin, 200 mM potassium phosphate buffer pH 6.0 and 2% glucose), buffered minimal methanol (BMM) medium (1.34% YNB, $4 \times 10^{-5}$% biotin, 200 mM potassium phosphate buffer pH 6.0) with 1% methanol (BMM2) or 5% methanol (BMM10). Antibiotic Zeocin (Invitrogen) was added to the media when required at a final concentration of 25 µg/mL for *E. coli* or 200 µg/mL for *P. pastoris* cultivations.

**Cloning and transformation of *P. pastoris***. De novo designed SARS-CoV-2 anti-RBD binder sequences *AHB2* (N-term-ELEEQVMHVLDQVSELAHELLHKLTGE ELERAAYFNWWATEMMLELIKSDDEREIREIEEEARRILEHLEELARK-C-term)[16] and *LCB1* (referred as LCB1v1.3)[17] (N-term-DKENILQKIYEIMKTLEQLGHAEAS MQVSDLIYEFMKQGDERLLEEAERLLEEVER-C-term) were codon optimised and fused to codon optimised CBD from cellobiohydrolase II *CH2* from *Trichoderma reesei* at the C-terminal end (N-term-ACSSVWGQCGGQNWSGPTCCASGSTC-VYSNDYYSQCL-C-term) or CBD from cellobiohydrolase *CEX* from *Cellulomonas fimi* at the N-term (N-term-PTSGPAGCQVLWGVNQWNTGFTANVTV KNTSSAPVDGWTLTFSFPSGQQVTQAWSSTVTQSGSAVTVRNAPWNGSIPAG GTAQFGFNGSHTGTNAAPTAFSLNGTPCTVG-C-term) gene sequences[18,19]. Linker sequence (N-term-GTPTPTPTPTGEF-C-term) was included between the anti-RBD binder sequence and CBD sequence[20]. Fused genes were ordered as gBlocks (IDT) and cloned by Gibson assembly into pD912 derived vector with α-mating-factor secretion signal according to protocols by Navone et al.[21], linearised with *Swa*I restriction enzyme and used to transform *P. pastoris* BG11 following standard electroporation protocol. *AHB2* and *LCB1* sequences without fusion to CBDs were also cloned into pD912 derived vector with α-factor secretion signal and transformed in *P. pastoris* for anti-RBD binders control experiments (construct and plasmids sequences can be found in Supplementary Data 1).

**Capture peptides expression in *P. pastoris***. Capture peptides AHB2-CH2, LCB1-CH2, AHB2-CEX and LCB1-CEX expression was conducted in 250 mL baffed shake flasks following standard expression conditions at 28 °C, 250 rpm. The culture was grown in 50 mL of BMD1 for 65 h following methanol induction with BMM10 and consecutive additions of pure methanol 1% final concentration until harvest at 132 h. Protein concentration in the culture supernatant was determined using the Bradford method[35]. The molecular weight of the expressed peptides was determined using Bolt 4-12% Bis-Tris 1.0 mm Mini Protein Gels or Novex 16% Tricine 1.0 mm Mini Protein gels. Endo H treatment and SDS-PAGE analysis confirmed the capture peptides and anti-RBD binders were not glycosylated during expression.

**Capture peptide binding assay to cellulosic materials**. Supernatant from *P. pastoris* cultivation expressing capture peptides AHB2-CH2, LCB1-CH2, AHB2-CEX and LCB1-CEX or anti-RBD AHB2 and LCB1 binders not fused to CBD, were

pH adjusted to 5, 6 or 7, filtered and incubated with 100% cotton (0.05 $cm^2$/mg surface area), 100% rayon (0.1 $cm^2$/mg surface area), Avicel PH101 (13 $cm^2$/mg) (Sigma)[31] at a concentration of 5 µg protein/mg of material in a final volume of 1.5 mL. Samples were incubated at 4 °C or room temperature and 150 RPM for 2, 4 or 16 h. Following incubation, samples were centrifuged for 5 min at 4 °C and 4500 RPM[20,33,34]. The concentration of total protein in the supernatant after centrifugation was determined by Bradford assay, and materials were stored at 4 °C for further testing. The percentage of binding of the capture peptides was calculated from the total protein in the solution before and after incubation. Removal controls of capture peptides to cellulosic materials were conducted by placing 30 mg of material in 400 uL deionised water, 50 mM potassium phosphate buffer, PBS or DMEM and incubated in a rotary platform at room temperature for 20 min. Bound cellulosic materials were also treated with 400 uL 1% SDS at 100 °C for 5 min. Each condition was repeated 5 times per sample consecutively, with protein concentration measured each time using Bradford Assay (detergent compatible for the samples containing 1% SDS)[35]. Data was processed using Prism software (GraphPad Prism 9.0).

**GFP-(SARS-CoV-2 RBD) cell-free protein expression**. Enhanced Green Fluorescent Protein GFP-(SARS-CoV-2 RBD) and GFP control were co-expressed in the *Leishmania tarentolae* translation competent extract (LTE) cell-free expression system, as previously described in[49], and updated in[50]. The DNA templates for N-terminal-eGFP (25 nM) tagged SARS-CoV-2 Spike protein RBD were added LTE for a final 50 µl or 100 µl LTE reaction mixture, incubated for 3 h at 25 °C, and stored at 4 °C prior to same-day binding experiments. Production of recombinant protein was tracked via the N-terminal eGFP fusion in a Tecan Spark plate reader.

**GFP-(SARS-CoV-2 RBD) binding assay to bioengineering textiles**. Bioengineered cotton with capture peptides AHB2-CH2 or LCB1-CH2 or unbound cotton was washed with 2% bovine serum albumin (BSA) (Sigma) to improve binding specificity for 20 min with slow rotation, centrifuged at 10,000 RPM for 5 min and incubated with 0.625 µg of GFP-(SARS-CoV-2 RBD) or GFP per 4 mg of cotton in a final volume of 100 µL of PBS for 1 h at room temperature. After incubation, samples were centrifuged at 10,000 RPM for 5 min and relative fluorescence units (RTU) of the supernatant measured at 485/510 nm. Binding to cotton was calculated as a percentage of the decrease in RFU before and after incubation with GFP-(SARS-CoV-2 RBD) or GFP. Data were processed using Prism software (GraphPad Prism 9.0).

**Confocal fluorescence microscopy**. Samples were washed with UHQ water and mounted onto glass coverslips. Z-stacked images were collected on a Nikon AR1 Confocal Microscope using the x20 objective and the 405 nm and 488 nm excitation wavelengths. Images were processed using Nikon NIS elements software.

**SARS-CoV-2 and HCoV-NL63 propagation and exposure assays**. SARS-CoV-2 (strain QLD02/2020, GISAID accession number EPI_ISL_407896) was obtained from Public Health Virology Laboratory, Queensland Health Forensic and Scientific Services. Virus stocks were produced in Vero E6 cells. Cells were infected with a stock vial of virus at a MOI of 0.1 PFU /mL in Dulbecco's (D)MEM/2% fetal calf serum/1% antibiotic–antimycotic (ThermoFisher) supplemented with 1 µg/mL tosyl phenylalanyl chloromethyl ketone (TPCK)-treated trypsin (Worthington Biochemical). Cell culture supernatant was collected after 4 days of incubation at 37 °C/5% $CO_2$, when a cytopathic effect was visible and 75% of cells were detached. Supernatant was clarified at 1500 × *g* for 5 min at 4 °C. HCoV-NL63 (Amsterdam-1 strain) was provided by Lia van den Hoek, University of Amsterdam, and propagated as previously described[51].

The titre of both virus stocks was calculated by TCID50/mL assay using a Spearman-Kaerber algorithm[51]. Viruses were used at a concentration of $1.36 \times 10^6$ particle-forming units (PFU) in DMEM (SARS-CoV-2) or OptiMEM (HCoV-NL63) for exposure to materials in this study.

Cotton pieces of 2 $cm^2$ were sterilised in the autoclave and filtered sterilised capture peptides AHB2-CH2 or LCB1-CH2 bound as previously described. For anti-RBD binder AHB2 and LCB1 controls *P. pastoris* culture supernatant was adjusted to pH 7, dialysed against PBS and filtered sterilised for viral binding experiments.

Pieces of bioengineered cotton or unbound cotton were placed in each well of a sterile 24-well-plate. The total mg of capture peptides attached to each cotton piece as calculated and the same amount was used as capture peptide AHB2-CH2 or LCB1-CH2 controls in solution in the 24-well-plate. SARS-CoV-2 or HCoV-NL63 alone (50 µL) was included as control on separate wells.

Wells containing bioengineered or unbound cotton were treated with 50 µL virus $1.36 \times 10^6$ TCID50/mL and incubated at room temperature for 1 h. After incubation, each well was washed with 150 µL DMEM or OptiMEM (for HCoV-NL63), for 10 min. Fifteen microliters of solution from each well were then placed in a sterile 96 well U-bottom plate of Vero E6 cells for SARS-CoV-2 or LLC-MK2 cells (ATCC, CCL-7) for HCV-NL63 for 10-fold virus serial dilutions in DMEM $10^{-1}$ to $10^{-4}$. cell plates were incubated for 4 days at 34 °C.

At day 4, liquid was removed from the plates and 50 mL of 80% methanol/water added as cell fixative for 1 h. Methanol was then removed and 50 mL of Crystal Violet (0.1%) in methanol (25%) stain was added for 15 min. The stain was then removed and the plates washed in deionised water. Cells were observed under the microscope and graded as dead or alive for Median Tissue Culture Infectious Dose (TCID50/mL) calculations using the Spearman Karber algorithm[51]. Data were processed using Prism software (GraphPad Prism 9.0).

**Statistical analysis**. All data are presented as mean ± SD for each group and analysed by ANOVA using GraphPad Prism. A *p*-value of less than 0.01 or 0.001 was considered statistically significant. The absorption data were fitted with Langmuir adsorption isotherm model using R studio 4.0.3. with PUPAIM package[52].

## Data availability

All data generated in this study are provided in Supplementary Data 1.

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

## Acknowledgements
The authors thank Dr Christopher Doropoulos (Commonwealth Scientific and Industrial Research Organisation, Australia) for his help on absorption data model fit. We also thank Professor David Baker from the University of Washington for kindly providing the antiviral peptide sequences prior to publication. This work was funded by the Centre for Agriculture and Bioeconomy and the Centre for a Waste Free World, Queensland University of Technology. We also acknowledge the Central Analytical Research Facility, operated and funded by Queensland University of Technology (QUT), and the financial support of the Australian Research Council Centre of Excellence in Synthetic Biology. Part of the figures was created with BioRender.com. L.N. is supported by QUT and the CSIRO Synthetic Biology Future Science Platform.

## Author contributions
L.N. designed and built genetic constructs, conducted experimental design, virus binding experiments, statistical analysis, and wrote the draft of the publication. K.M. carried out the expression of constructs and binding experiments. K.S. designed and carried out virus binding experiments. W.J. conducted cell-free expression of fluorescent constructs. C.C. performed confocal microscopy. T.M. reviewed and edited the manuscript for publication. R.S. contributed to the experimental design and edited the draft for publication. R.S., K.S. and L.N. developed the concept and obtained the funding. All authors reviewed the paper.

## Competing interests
The authors declare no competing interests.
