## [Peer Review File · Communications Materials]

Web links to the author's journal account have been redacted from the decision letters as indicated to maintain confidentiality.

14th Apr 22

Dear Dr Navone,

Thank you for submitting your manuscript, "Bioengineered textiles that capture SARS-CoV-2 viral particles.", to Communications Materials. It has now been seen by 2 referees, whose comments are appended below. You will see that while they find your work of potential interest, they have raised substantial concerns that must be addressed. In light of these comments, we cannot accept the manuscript for publication, but are interested in considering a revised version that addresses these serious concerns.

You will see that Reviewer 1 raises a number of potential limitations of this mask, and requests that additional experiments be performed. In particular, they are requesting data to understand protein stability, pore size, breathability and time-dependent virus capture. They also raise some doubts regarding mask performance, which requires additional discussion.

We hope you will find the referees' comments useful as you decide how to proceed. Should further experimental data and analysis allow you to address these criticisms, we would be happy to look at a substantially revised manuscript. However, please bear in mind that we will be reluctant to approach the referees again in the absence of major revisions. If the revision process takes significantly longer than three months, we will be happy to reconsider your paper at a later date, as long as nothing similar has been accepted for publication at Communications Materials or published elsewhere in the meantime.

When submitting your revised manuscript, please include the following:

-A response letter with a point-by-point reply to each of the referee comments and a description of changes made. Please include the complete referee report in the response letter. Please note that the response letter must be separate to the cover letter to the editors.

-A marked-up version of the manuscript with all changes to the text in a different colored font. Please do not include tracked changes or comments. Please select the file type 'Revised Manuscript - Marked Up' when uploading the manuscript file to our online system.

-A clean version of the manuscript. Please select the file type 'Article File'.

-An updated <https://www.nature.com/documents/nr-editorial-policy-checklist.zip> Editorial Policy checklist, uploaded as a 'Related Manuscript File' type. This checklist is to ensure your paper complies with all relevant editorial policies. If needed, please revise your manuscript in response to these points. Please note that this form is a dynamic 'smart pdf' and must therefore be downloaded and completed in Adobe Reader. Clicking this link will download a zip file containing the pdf.

Please use the following link to submit your revised manuscript files:

[link redacted]

We understand that due to the current global situation, the time required for revision may be longer than usual. We would appreciate it if you could keep us informed about an estimated timescale for resubmission, to facilitate our planning. Of course, if you are unable to estimate, we are happy to accommodate necessary extensions nevertheless.

Please do not hesitate to contact me if you have any questions or would like to discuss the required revisions further. Thank you for the opportunity to review your work.

Best regards,

John Plummer, PhD
Chief Editor
orcid.org/0000-0003-4824-8497
Communications Materials

Reviewers' comments:

Reviewer #1 (Remarks to the Author):

In this study, the authors designed the hybrid protein, which has the high binding capability to SARS-CoV-2 and cellulose, to develop the bioengineered protective textiles. Hybrid protein includes a bidirectional promoter and linker to produce the protein massively. Practical methods confirmed the strong and stable binding affinity of CH2 and CEX to cotton. In addition, optical observation and cell culture tests were performed systematically to show the specific binding of hybrid protein to SARS-CoV-2. However, the novelty of their work should be explained more and some points have to be fixed to improve the quality of this work as suggested following.

1. What is the benefit of this technique compared to KN95, N95, and KN94? Those masks have enough capability to block the virus spread. The cotton mask is not efficient to block the virus contact due to high airflow. The efficiency of a mask applied to this technique is still low (less than 60%). Due to that there is needed the stability of the hybrid protein data (how long? As shelf life of mask or textile product is more than few years!) while attached in cotton and need to provide the data for the stability of the mask itself by employing the protein
2. Viruses can stick to cotton as the authors mentioned in the introduction (fomite transmission). How can an author distinguish between protein binding and non-specific binding of the virus on cotton? If you capture the virus, then what is the next step?
3. If the author repeats the coating process to the cotton, can the coverage of hybrid protein on the mask increase? And various binding proteins for the various virus can be attached to cotton simultaneously?
4. In general, mask such as NK95, and KN94 blocks the virus by controlling the pore size. Compared

to the droplet size from coughing, what is the size of the pore of the cotton mask is used in this study and blocking efficiency of the droplet?

5. Breathability is the main parameter for the mask. What is the breathability of cotton masks after coating hybrid protein? Authors should add this study to the main text.

6. The analysis of the amount of captured virus as a function of time-lapse is important to show the efficiency of hybrid protein-coated mask.

7. The technique in this study might be good to apply to mass production air filtration once you confirm the stability and efficacy of capturing virus. Are there any obstacles to apply this technique to mass air filtration in terms of cost and process and there are solutions?

8. PPE that we used widely is composed of a synthetic polymer such as Polyphenylene Ether (PPE). Author should explain the advantage of using this technique.

9. For the usage of hybrid protein-coated masks, how long do they keep their function in various conditions such as temperature, humidity, and light? This is the related question to above stability of the mask.

Minor comments

- Reference format is not correct (Pg 5, line 122)

Reviewer #2 (Remarks to the Author):

The manuscript demonstrates a cloth that can bind SARS-CoV-2. I think that the results are timely and well proven. I recommend only minor changes to this manuscript.

Introduction

The introduction is clear. I suggest that the authors add a small schematic showing the proposed linkage of the spike protein to the cellulose with the links between. Figure 5 in the discussion is a possibility for this

Results

Figure 1. Surgical facemasks are not made from cellulose, so this picture is an unfortunate choice.

The authors made a convincing argument that the construct that binds to the spike protein can be bound to cellulose. They also showed that it was resistant to rinsing by water. It is unfortunate that the binding was insufficient to resist cleaning with SDS so the materials will not be able to be laundered and are thus disposable only. They also made a convincing argument based on GFP reporting that the construct binds the spike protein and also that SARS-CoV-2 binding to the modified cotton is superior to regular cotton.

It is purely a matter of wording but I think the authors could improve lines 245-248. This is the key result of the manuscript and it is written in a complex manner. Also "unbound cotton" is ambiguous. Unmodified cotton would be better.

Discussion:

Lines 298-233 are just a long reiteration of the results and are not really a discussion. The only real discussion was lines 334-343. I would prefer a short "Conclusions" where the conclusions are described succinctly in a couple of sentences separately from discussion.

The authors refer to copper and silver-based antimicrobials: "these antiviral coatings expose

individuals to heavy metals that may lead to health issues,” It would be reasonable then to discuss possible health issues of the new binding proteins.

Also, because the binding proteins are not strongly held, it would be reasonable to discuss abrasion resistance.

References

1. The introduction discusses the SARS-CoV-2 virus can be transmitted via fomites. There is a recent paper by Behzadinasab, S., Chin, A.W.H. et al. *Sci Rep* 11, 22868 (2021) showing that the virus transmits from fomites to skin. This could be added to the introduction.
2. On page 5, line 122, the two references (by Chen et al and Taylor et al) need to be added to the reference section.

We thank the reviewers and editor for the positive feedback. All issues raised are addressed below.

Reviewer #1 (Remarks to the Author):

In this study, the authors designed the hybrid protein, which has the high binding capability to SARS-CoV-2 and cellulose, to develop the bioengineered protective textiles. Hybrid protein includes a bidirectional promoter and linker to produce the protein massively. Practical methods confirmed the strong and stable binding affinity of CH2 and CEX to cotton. In addition, optical observation and cell culture tests were performed systematically to show the specific binding of hybrid protein to SARS-CoV-2. However, the novelty of their work should be explained more and some points have to be fixed to improve the quality of this work as suggested following.

We want to thank reviewer #1 for taking the time to review our manuscript and we would like to clarify several points the reviewer has conveyed. As a broad comment, we believe the main purpose of our research has not been well understood by reviewer #1, and that they have focused on one potential application. We want to clarify that the bioengineered cotton we developed here is not specifically targeted for masks. Nor did we intend in our work to improve any previous developed masks like KN95. We strongly believe the bioengineered textile is a novel material that can be applied in many forms to protect the population from the spread of viruses. Examples, as described in the manuscript are PPE, ventilation systems to trap and neutralise airborne viral particles, clothing and furnishings to limit fomite transmission in hospitals, age care facilities and public transport. We used a novel synthetic biology approach to develop a new material's technology presenting a new concept for community protection against pathogens. We apologise if the manuscript has focussed the attention on the use for masks and we can rephrase accordingly if the editor believes it is needed for improvement. We have modified Fig. 1 to include other applications and avoid confusion.

Furthermore, some of the questions and concerns raised by reviewer #1 were already discussed in the original manuscript, which suggests that the reviewer might have missed some parts. We indicate the lines in each response below where we had already presented information relevant to the comments or questions of the reviewer.

1. What is the benefit of this technique compared to KN95, N95, and KN94? Those masks have enough capability to block the virus spread. The cotton mask is not efficient to block the virus contact due to high airflow. The efficiency of a mask applied to this technique is still low (less than 60%). Due to that there is needed the stability of the hybrid protein data (how long? As shelf life of mask or textile product is more than few years!) while attached in cotton and need to provide the data for the stability of the mask itself by employing the protein.

We want to clarify that the main aim of the work was not intended to the improvement of currently available masks or develop new masks. However, with the intent to reply to the reviewer's comment, the benefit of the material is that it can be reusable compared to N95, etc. We have proved the material can be easily washed with detergent to remove viral particles attached to the hybrid protein in the surface of the cotton. The detergent completely removes the bound hybrid protein from the cotton, removing any attached virus as well. Furthermore, as presented in the results section of the manuscript, the hybrid protein can be easily re-attached to the cotton, making the bioengineered textile reusable by wash and re-load. These results can be found in results section *Attachment of capture peptides to textiles* and are later discussed in lines 316-322.

We are unsure of what the reviewer is referring to with 60% efficiency of cotton masks. Regarding stability, the hybrid capture peptides stay bound to cotton for up to 4 weeks and still binds GFP-SARS-CoV-2-RBD, suggesting that it could most likely still bind SARS-CoV-2 virus after the same period of time. Regarding shelf life, we did not include experiments to test this because we envisioned the application of our technology on a daily or weekly basis. For example, the bioengineered cotton could be used for bedding in hospitals or age care facilities that could be washed and re-loaded with capture peptides on a daily or weekly basis. If the bioengineered cotton is used in reusable cloth masks the same concept applies, use then wash and reload with capture peptide. The production of the capture peptides in yeast, as shown in the manuscript, supports the idea of low-cost manufacture which makes a reloading approach economically feasible.

2. Viruses can stick to cotton as the authors mentioned in the introduction (fomite transmission). How can an author distinguish between protein binding and non-specific binding of the virus on cotton? If you capture the virus, then what is the next step?

We have tested and compared binding of SARS-CoV-2 virus to cotton alone (with no bound capture peptide) and bioengineered cotton (bound with capture peptide), showing that cotton alone does not bind virus significantly in the experimental conditions tested. These results can be found in section *Bioengineered cotton neutralises SARS-CoV-2 infection* Fig 4.

As mentioned previously, once the virus is bound it can be washed off with detergent (Supplementary Fig 2f), however, certain chemicals (like quaternary ammonium chloride compounds or other non-toxic nanocoatings) could be added to achieve both capture and kill, but this was outside the scope of this work. The virus could be fully inactivated during washing procedures after use. We discuss this idea in line 334-337 of the manuscript.

3. If the author repeats the coating process to the cotton, can the coverage of hybrid protein on the mask increase? And various binding proteins for the various virus can be attached to cotton simultaneously?

Theoretically, we do not expect more hybrid protein to be bound on a second binding reaction to the same piece of cotton. This derives from the Langmuir Isotherm Absorption model of binding. We show these results in Supplementary Fig. 2.

The technology allows to use a combination of capture peptides that bind different viruses to target different pathogens on the same material. We have discussed this idea in lines 303-307 of the manuscript.

4. In general, mask such as NK95, and KN94 blocks the virus by controlling the pore size. Compared to the droplet size from coughing, what is the size of the pore of the cotton mask is used in this study and blocking efficiency of the droplet?

We did not test pore size of the cotton used. By including cotton alone (unbound to capture peptide) in the experiments we normalised for specificities of the material and have a direct comparison to the same material bound to capture hybrid peptide. We want to emphasize again that the technology is not only developed for the use for masks. The pore size is not strictly important if the material is to be used in beddings or furnishing for example. Because the work was not targeted to a specific application we did not conduct experiments that were exclusively needed for application in

PPE or filtration systems or any of the suggested potential applications of the technology. Further work would be required to test specific applications in real world settings. At this point factors such as droplet size will be important.

5. Breathability is the main parameter for the mask. What is the breathability of cotton masks after coating hybrid protein? Authors should add this study to the main text.

We do not believe the proposed study is necessary for the work presented here. Again, we did not develop the technology targeting the use of mask specifically. We understand that breathability is important in the development of materials for the use masks, however, this work was not conducted with the aim of developing improved masks but with the intention of creating an innovative textile with broad applicability. If the bioengineered textile is used for the manufacture of masks in the future, then studies of breathability would need to be conducted in that case, as well as many other studies depending on the particular application.

6. The analysis of the amount of captured virus as a function of time-lapse is important to show the efficiency of hybrid protein-coated mask.

Working with highly transmissible pathogens, like SARS-CoV-2, is a challenging process requiring a lot of training and resources and exposing the workers to potential infection. For the purpose of showing the efficacy of binding of the virus to the bioengineered textile we did not consider a time-lapse experiment but chose a time (1 h) that most likely exceeds real-world scenarios of exposure to the virus. Time-lapse experiments also imply staff being exposed to the virus for extended time, for safety reasons we preferred to limit as much as possible the exposure to the pathogen.

We want to clarify again that did not develop a 'protein-coated mask' in this work, but a bioengineered textile.

7. The technique in this study might be good to apply to mass production air filtration once you confirm the stability and efficacy of capturing virus. Are there any obstacles to apply this technique to mass air filtration in terms of cost and process and there are solutions?

We have mentioned this potential application in line 298 and Fig. 5 of the manuscript. We do not foresee any major obstacles to applying the technology to air filtration systems in terms of cost or process, but we have not investigated this application in detail. Other studies targeting this application should be conducted by specialists in the area to answer these specific questions.

Stability and capturing capability of the technology should be tested in the specific conditions of filtering systems if used for this application. The same way we did not test breathability of the bioengineered textile in this work because we are not developing masks, we also did not test stability of other properties in air filtering systems.

8. PPE that we used widely is composed of a synthetic polymer such as Polyphenylene Ether (PPE). Author should explain the advantage of using this technique.

We do not fully understand the reviewer's comment. If the capture peptides were to be applied in facemasks, it would be to cloth masks made of cellulosic fibers, or the binding domain would need to be exchanged to a polyphenylene ether binding domain for attachment to the material.

9. For the usage of hybrid protein-coated masks, how long do they keep their function in various conditions such as temperature, humidity, and light? This is the related question to above stability of the mask.

We have already addressed this comment, please refer to our response of question 1.

Minor comments

- Reference format is not correct (Pg 5, line 122)

We have fixed the reference in the new version of the manuscript.

Reviewer #2 (Remarks to the Author):

The manuscript demonstrates a cloth that can bind SARS-CoV-2. I think that the results are timely and well proven. I recommend only minor changes to this manuscript.

We thank reviewer #2 for the positive feedback and constructive comments on the manuscript.

Introduction

The introduction is clear. I suggest that the authors add a small schematic showing the proposed linkage of the spike protein to the cellulose with the links between. Figure 5 in the discussion is a possibility for this.

We have referred to the reader to Fig. 1 in the introduction which includes a schematic representation of the bioengineered textile. We were unsure about incorporation an additional figure in the manuscript. We believe Fig. 5 fits really well in the discussion and we decided not to move it to introduction. If the editor still believes an additional diagram is required in the introduction we can include one.

Results

Figure 1. Surgical facemasks are not made from cellulose, so this picture is an unfortunate choice.

The facemask drawing of figure 1 was not intended to be a surgical facemask but to represent a cloth mask. We have modified the drawings to avoid confusion and misinterpretation of the bioengineered textile to be targeted for facemasks and included other applications to communicate a broader idea.

The authors made a convincing argument that the construct that binds to the spike protein can be bound to cellulose. They also showed that it was resistant to rinsing by water. It is unfortunate that the binding was insufficient to resist cleaning with SDS so the materials will not be able to be laundered and are thus disposable only. They also made a convincing argument based on GFP reporting that the construct binds the spike protein and also that SARS-CoV-2 binding to the

modified cotton is superior to regular cotton.

We thank the reviewer for the positive comment and would like to point out that despite the capture peptides are removed with laundry detergent they could easily be reapplied, so we do not think the material is disposable. We discuss this idea in line 321-323.

It is purely a matter of wording but I think the authors could improve lines 245-248. This is the key result of the manuscript and it is written in a complex manner. Also “unbound cotton” is ambiguous. Unmodified cotton would be better.

We thank the reviewer for the comment and have re-worded lines 245-248 in the new version of the manuscript. We understand that unbound could be ambiguous in some cases but we do not think that ‘unmodified cotton’ is strictly correct. We are not modifying the fibres or the structure of cotton therefore, we prefer to use alternatives like ‘unbound’ and avoid the use of ‘modified’.

Discussion:

Lines 298-333 are just a long reiteration of the results and are not really a discussion. The only real discussion was lines 334-343. I would prefer a short “Conclusions” where the conclusions are described succinctly in a couple of sentences separately from discussion.

We thank the reviewer for the comment, we do not agree that lines 298-333 are a reiteration of results. We do refer back to results to set the starting point for discussion and guide the reader but we do not believe that results are presented in a reiterative manner. We have included a short conclusion from line 339-351.

The authors refer to copper and silver-based antimicrobials: “these antiviral coatings expose individuals to heavy metals that may lead to health issues,” It would be reasonable then to discuss possible health issues of the new binding proteins.

We do not foresee health issues with the capture peptides. The inhibitory peptides (anti-RBD binders) were developed as therapeutic and prophylactics and proved to be safe in cell and animal models (see references Case et. al, 2021 and Cao et. al, 2020). We wouldn’t expect the incorporation of the cellulose binding domain to the anti-RBD binders to have a negative effect on health. We briefly comment this in line 331.

Also, because the binding proteins are not strongly held, it would be reasonable to discuss abrasion resistance.

The capture peptides bind strongly to cotton with very low value K_D values as shown on Supplementary Fig. 2 and discussed in line 149-153. Furthermore, the inhibitory peptides (anti-RBD binders) also bind strongly to SARS-CoV-2 RBD as shown in previous publications (see references Case et. al, 2021 and Cao et. al, 2020 and line 67 in the manuscript). However, abrasion would need to be considered depending on the application of the bioengineered cotton. Particularly if used for bedding and furnishings. We have added a comment regarding this in line 328-330.

References

1. The introduction discusses the SARS-CoV-2 virus can be transmitted via fomites. There is a recent paper by Behzadinasab, S., Chin, A.W.H. et al. Sci Rep 11, 22868 (2021) showing that the virus transmits from fomites to skin. This could be added to the introduction.
2. On page 5, line 122, the two references (by Chen et al and Taylor et al) need to be added to the reference section.

We have incorporated the suggested reference and fixed referencing on line 122.

5th Jul 22

Dear Dr Navone,

Your manuscript titled "Bioengineered textiles that capture SARS-CoV-2 viral particles." has now been seen again by Reviewer 1, whose comments appear below. In light of their advice I am delighted to say that we are happy, in principle, to publish a suitably revised version in Communications Materials under the open access CC BY license (Creative Commons Attribution v4.0 International License).

We therefore invite you to edit your manuscript to comply with our journal policies and formatting style in order to maximise the accessibility and therefore the impact of your work.

EDITORIAL REQUESTS

* Your manuscript should comply with our policies and format requirements, detailed in our style and formatting guide (<https://www.nature.com/documents/commsj-phys-style-formatting-guide-accept.pdf>).

* Please edit your manuscript according to the editorial requests in the attached table, and outline revisions made in the right hand column. If you have any questions or concerns about any of our requests, please do not hesitate to contact me. It is important that each request be addressed in order to avoid delays in accepting your manuscript. Please upload the completed table with your manuscript files.

* The editorial requests table also includes a full list of the files that must be provided upon resubmission. Please upload your files according to this table.

* An updated editorial policy checklist that verifies compliance with all required editorial policies must be completed and uploaded with the revised manuscript. All points on the policy checklist must be addressed; if needed, please revise your manuscript in response to these points. Please note that this form is a dynamic 'smart pdf' and must therefore be downloaded and completed in Adobe Reader. Clicking this link will download a zip file containing the pdf.

OPEN ACCESS

Communications Materials is a fully open access journal. Articles are made freely accessible on publication under a [CC BY](http://creativecommons.org/licenses/by/4.0) license (Creative Commons Attribution 4.0 International License). This license allows maximum dissemination and re-use of open access materials and is preferred by many research funding bodies.

For further information about article processing charges, open access funding, and advice and support from Nature Research, please visit <https://www.nature.com/commsmat/about/open-access>

RESUBMISSION

At acceptance, you will be provided with instructions for completing this CC BY license on behalf of

all authors. This grants us the necessary permissions to publish your paper. Additionally, you will be asked to declare that all required third party permissions have been obtained, and to provide billing information in order to pay the article-processing charge (APC).

Please use the following link to submit your revised files:

[link redacted]

We hope to hear from you within two weeks; please let us know if the process may take longer.

Best regards,

John Plummer, PhD
Chief Editor
orcid.org/0000-0003-4824-8497
Communications Materials

REVIEWERS' COMMENTS:

Reviewer #1 (Remarks to the Author):

The authors have nicely revised the manuscript and clarified the novelty and impact of the manuscript, with which I agree. I recommend that it is now suitable for publication in Communications Materials.